# Insights into the Protein Differentiation Mechanism between Jinhua Fatty Ham and Lean Ham through Label-Free Proteomics

**DOI:** 10.3390/foods12234348

**Published:** 2023-12-01

**Authors:** Qicheng Huang, Ruoyu Xie, Xiaoli Wu, Ke Zhao, Huanhuan Li, Honggang Tang, Hongying Du, Xinyan Peng, Lihong Chen, Jin Zhang

**Affiliations:** 1State Key Laboratory for Managing Biotic and Chemical Threats to the Quality and Safety of Agro-Products, Institute of Food Science, Zhejiang Academy of Agricultural Sciences, Hangzhou 310021, China; hqc605322418@outlook.com (Q.H.); xieruoyu2023@126.com (R.X.); 18277970590@163.com (X.W.); kzhao@snnu.edu.cn (K.Z.); huanhuanlee325@163.com (H.L.); zaastang@163.com (H.T.); cwc528@163.com (L.C.); 2Department of Food Science and Engineering, College of Light Industry and Food Engineering, Nanjing Forestry University, Nanjing 210037, China; hydu@njfu.edu.cn; 3College of Life Science, Yantai University, Yantai 264005, China; pengxinyan2006@ytu.edu.cn

**Keywords:** fatty ham, lean ham, Jinhua ham, label-free proteomics, HPLC-MS/MS, metabolic mechanism

## Abstract

Jinhua lean ham (LH), a dry-cured ham made from the defatted hind legs of pigs, has become increasingly popular among consumers with health concerns. However, the influence of fat removal on the quality of Jinhua ham is still not fully understood. Therefore, a label-free proteomics strategy was used to explore the protein differential profile between Jinhua fatty ham (FH) and lean ham (LH). Results showed that 179 differential proteins (DPs) were detected, including 82 up-regulated and 97 down-regulated DPs in LH vs. FH, among which actin, myosin, tropomyosin, aspartate aminotransferase, pyruvate carboxylase, and glucose-6-phosphate isomerase were considered the key DPs. GO analysis suggested that DPs were mainly involved in binding, catalytic activity, cellular process, and metabolic process, among which catalytic activity was significantly up-regulated in LH. Moreover, the main KEGG-enriched pathways of FH focused on glycogen metabolism, mainly including the TCA cycle, pyruvate metabolism, and glycolysis/gluconeogenesis. However, amino acid metabolism and oxidative phosphorylation were the main metabolic pathways in LH. From the protein differentiation perspective, fat removal significantly promoted protein degradation, amino acid metabolism, and the oxidative phosphorylation process. These findings could help us to understand the effects of fat removal on the nutritional metabolism of Jinhua hams and provide theoretical supports for developing healthier low-fat meat products.

## 1. Introduction

Jinhua ham is a high-quality dry-cured meat product with a history of over 1200 years. Because of its bright color, unique flavor, good taste, and high nutritive value, Jinhua ham has been listed in the “China National Geographic Indication Products”. As a traditional ethnic food in China, Jinhua ham is widely popular among consumers as a flavor enhancer, umami improver, or nutrient fortifier in dishes. Jinhua ham is made from the fresh hind legs of pigs through a process of salting, dry-ripening, and post-ripening, which usually takes months to ferment and decompose nutrients, making it easier to absorb. During the long ripening process, the degradation of protein, fat, and glycogen in raw ham accumulated rich amino acids, fatty acids, and pyruvate [1], which further formed the special flavor of dry-cured ham through β-oxidation, deamination, Strecker degradation, and the Millard reaction [2].

Fat is a double-edged sword in the ripening of dry-cured hams, which can not only provide rich flavor precursors, but also produce bad flavor through excessive oxidation. In recent years, consumers have placed an increased emphasis on healthy diet, since the excessive consumption of animal fats, mainly saturated fats, can be associated with a higher risk of obesity, arteriosclerosis, and coronary heart disease [3]. Thus, lean ham (LH), prepared from pig legs with subcutaneous fat tissues removed, is gaining popularity among consumers, especially in people with health concerns. Our previous study [4] found that fat removal could significantly affect the volatile flavor characteristics of dry-cured hams. Specifically, LH has a higher content of total volatile alcohols and acids, while fatty ham (FH, without removing fats) generally has richer volatile aldehydes, ketones, esters, and heterocyclic and sulfur-containing compounds [4]. Protein and its degradation products, such as peptides and amino acids, occupy the highest content of nutrients in dry-cured hams. Moreover, proteolysis is one of the most important biochemical reactions across the whole processing of dry-cured hams, which is considered as a main factor affecting taste and aroma [5]. In addition, the degradation and oxidation products of fats can also participate in metabolism pathways of amino acids through the tricarboxylic acid (TCA) cycle during the ripening of dry-cured hams. Hence, it is of great significance to investigate the effect of fat removal or deficiency on the protein-related metabolism of dry-cured hams. However, this effect and its associated mechanism are still rarely studied and not fully understood.

Proteomics is a scientific method for the analysis and identification of large-scale proteins based on mass spectrometry [6]. Meanwhile, label-free proteomics has been recognized as a simple, reliable, versatile, and cost-effective strategy in the identification of biomarkers for the quality traits of meat products [7]. Previous studies have reported several applications of label-free proteomics in dry-cured ham investigations, such as screening marker proteins that cause bitterness and adhesion of defective hams [8], exploring the mechanism of flavor improvement by ultrasonic treatment for defective hams [9], and analyzing the difference in protein profiles between hams with modern and traditional processing [10]. Therefore, the purpose of this study was (i) to compare the protein differentiation between Jinhua FH and LH, and (ii) to explore the underlying mechanism of fat removal affecting the nutritional metabolism of Jinhua hams from the protein differentiation perspective. The information obtained from this study could provide theoretical supports for developing healthier dry-cured meat products.

## 2. Materials and Methods

### 2.1. Processing of Fatty Ham and Lean Ham

The Jinhua hams used in this study were manufactured at Jinhua Jinnian Ham Co., Ltd. (Jinhua, China) following the procedures of Zhang et al. [4] with minor modifications. Specifically, for Jinhua fatty ham (FH), three hind legs were marinated with 0.014% NaNO_2_ and 10% NaCl for 75 days, followed by soak cleaning for 1 day and sun-drying for 1 day. After drying, the hind legs were dehydrated for 7 days, sun-dried for another 1 day, and then dry-ripened for 180 days. During the dry-ripening process, the temperature gradually increased from 5 °C to 35 °C, and the relative humidity decreased from 85% to 65%. Afterward, the hind legs were further post-ripened at 25 °C for about 30 days to obtain the final Jinhua FH, which was terminated when the weight loss of each hind leg reached 40% of its initial weight. For Jinhua lean ham (LH), skin and fat tissues were firstly removed from the other three hind legs, and then followed by the same process as Jinhua FH. About 20 g of the internal samples was taken from the central part of the *biceps femoris* muscle for each FH or LH (about 3–4 cm depth). All samples were vacuum-packed and frozen at 80 °C for further analysis.

### 2.2. Protein Extraction and Digestion

SDT buffer (4% SDS, 100 mM Tris-HCl, 1 mM DTT, and pH = 7.6) was used for the protein extraction of each ham sample. The amount of protein was quantified with the BCA Protein Assay Kit (Bio-Rad, Hercules, CA, USA). Protein digestion was performed according to a filter-aided sample preparation (FASP) procedure described by Wiśniewski et al. [11]. The peptides from the protein digestion of each sample were desalted on C18 Cartridges (Empore^TM^ SPE Cartridges C18 (standard density), bed I.D. 7 mm, volume 3 mL, Sigma, Darmstadt, Germany), concentrated by vacuum centrifugation and reconstituted in 40 µL 0.1% (*v*/*v*) formic acid.

### 2.3. HPLC-MS/MS Analysis

A HPLC-MS/MS Easy-nLC system was used to separate sample peptide fragments at a nanoliter flow rate. Buffer solution A was 0.1% formic acid aqueous solution and buffer solution B was 0.1% formic acid acetonitrile aqueous solution (84% acetonitrile). The column was balanced with 95% solution A, and samples were loaded from an automatic injector into a trap column (Acclaim PepMap100, 100 μm × 2 cm, nanoViper C18, Thermo Fisher Scientific, Waltham, MA, USA), and then passed into an analytical column (EASY column, 10 cm, ID75 μm, 3 μm, C18-A2, Thermo Fisher Scientific, Waltham, MA, USA) for separation at a flow rate of 300 nL/min.

The separated samples were analyzed using a Q-Exactive mass spectrometer (MS), which was operated in a positive ion mode. The MS data were acquired with a data-dependent top 10 method by dynamically choosing the most abundant precursor ions from the survey scan (300–1800 *m*/*z*) for high energy collision dissociation (HCD) fragmentation. The automatic gain control (AGC) target was set to 3 × 10^6^, the maximum inject time was 10 ms, and the dynamic exclusion duration was 40.0 s. Survey scans were acquired at a resolution of 70,000 at *m*/*z* 200, the resolution for HCD spectra was 17,500 at *m*/*z* 200, and the isolation width was set to 2 *m*/*z*. The normalized collision energy was 30 eV, while the underfill ratio, specifying the minimum percentage of the target value likely to be reached at maximum fill time, was defined as 0.1%. The instrument was run with peptide recognition mode enabled.

### 2.4. Protein Identification and Quantitative Analysis

MaxQuant software (1.5.3.17) was applied for identification and quantitative analysis against the MS raw data and Uniport pig database with the following parameters. Trypsin was used as the protein-cleaving enzyme, and two missed cleavages were the upper limit. Carbamidomethyl (C) and methionine oxidation (M) were set as variable modifications. The main research, first research, and MS/MS tolerance were 6 ppm, 20 ppm, and 20 ppm, respectively. A significant threshold of *p* < 0.05 and false discovery rate (FDR) ≤ 0.01 were set as the qualitative analysis criterion. Meanwhile, the quantitative analysis was performed based on the label-free quantitation (LFQ) intensity of each peak. For LH vs. FH comparison, fold-change (FC) > 1.5 or <0.667 and *p* < 0.05 were set as the thresholds to identify the proteins with differential abundances. Moreover, the proteins detected in ≥2 samples of one group but not detected in any sample of the other group were recognized as different proteins with a consistent presence/absence.

### 2.5. Bioinformatic Analysis

Above all, the LFQ intensity of proteins with differential abundances was normalized to the interval from −1 to 1. A cluster analysis of differential proteins was processed by Complexheatmap R package V3.4 with two dimensions (sample and protein expression) classified and the hierarchical clustering heatmaps generated. Subcellular localization was predicted by CELLO (http://cello.life.nctu.edu.tw/, accessed on 22 October 2022), which is a multi-class support vector machine classification system.

GO is a standardized functional classification system providing a dynamically updated standardized vocabulary for property descriptions of genes and gene products in an organism [12]. The protein sequences of identified DPs were locally searched using the NCBI BLAST+ NCBI client software and InterProScan to obtain homologue sequences. Then, the gene ontology (GO) terms were mapped, and sequences were annotated using the software program Blast2GO. Moreover, the GO annotation results were plotted using R scripts. Afterward, the DPs were blasted against the online Kyoto Encyclopedia of Genes and Genomes (KEGG) database (http://geneontology.org/, accessed on 22 October 2022) to retrieve their KEGG orthology identifications and then were mapped to metabolic pathways. Subsequently, the enrichment analysis was performed using Fisher’ exact test, considering the whole quantified proteins as a background dataset. Benjamini–Hochberg correction for multiple testing was further used to adjust derived *p*-values. And only functional categories and pathways with *p* < 0.05 were confirmed as significant.

### 2.6. Statistical Analysis

All experiments were performed in biological triplicate with results shown as means ± standard deviation. Tables were made using the Microsoft Excel 2019 software, while figures were drawn using Origin 2021 (OriginLab, Northampton, MA, USA) and Microsoft PowerPoint 2019. The significance of differences among the mean values was determined by one-way analysis of variance (ANOVA) and Waller–Duncan’s test using SPSS V22.0 (SPSS software Inc., Chicago, IL, USA). The difference was confirmed to be statistically significant when *p* < 0.05.

## 3. Results

### 3.1. Identification of Differential Proteins (DPs)

A Venn diagram was used to analyze the differences in protein profiles within and between groups of FH and LH, and results are shown in Figure 1a–c. As shown in Figure 1a,b, a total of 1230 and 1275 proteins were identified in FH and LH samples, respectively. As illustrated in Figure 1c, 1305 proteins in total were found in FH and LH, including 1200 overlapping proteins and 105 different proteins. Specifically, FH and LH had 30 and 75 different proteins, respectively.

According to the report of Zhou et al. [5], the overlapping proteins with differential abundances (FC > 1.5 or <0.667 and *p* < 0.05) were also recognized as DPs between FH and LH, except for the different proteins with a consistent presence/absence. The numbers of these two kinds of DPs are shown in Figure 2. On the one hand, it is illustrated in Figure 2a that the abundances of 66 and 47 overlapping proteins are significantly up-regulated >50% in FH and LH, respectively. Furthermore, two of these proteins (annexin and 1 uncharacterized protein) were up-regulated > 9 times in FH, whereas five proteins (adiponectin, protein-serine/threonine kinase, and three uncharacterized proteins) were up-regulated > 9 times in LH. On the other hand, as displayed in Figure 2b, a total of 66 different proteins with consistent presence/absence were identified, including 50 proteins up-regulated in FH and 16 proteins up-regulated in LH, respectively. Both of these two types of DPs were further studied for the analysis of bioinformatic function.

### 3.2. Characterization of Proteins with Differential Abundances

A volcanic map of proteins with differential abundances in LH vs. LH comparison is shown in Figure 3. Information about characterized proteins among them, including protein source, gene source, intensities, FC, and *p* values, is shown in Table 1. In combination with Figure 3 and Table 1, it is clear that many characterized proteins with significantly up-regulated abundances in FH were associated with nutrient metabolism, such as TCA cycle and pyruvate metabolism (pyruvate carboxylase), glycogen metabolism (glucose-6-phosphate isomerase, fructose-bisphosphatase, and ATP-dependent 6-phosphofructokinase), and nucleic acid metabolism (deoxyribonuclease and aminoacyl-tRNA hydrolase). In addition, the results also showed that some muscle structural proteins (tropomyosin and actin) and ribosomal protein were up-regulated in FH. Actin is a key protein of the contracting unit of muscle fibrils, which is formed by the crossbridge within actin and myosin [13]. Tropomyosin is involved in the regulation of myosin and actin filament assembly [9].

On the other hand, myosin, calcium-transporting ATPase, aspartate aminotransferase, protein kinase domain-containing protein, nucleoside diphosphate kinase, calcium/calmodulin-dependent protein kinase, and numerous other characterized proteins were significantly up-regulated in LH vs. FH, among which many proteins were enzymes related to protein degradation. Calpain mainly degrades intermediate fibrin and structural protein. Early proteolysis of structural proteins and cytoskeletal anchoring complexes by the calpain system is related to the texture quality of pork [14]. It has been reported that the degradation of sarcoplasmic and myofibrillar proteins largely determines the texture and taste characteristics of dry-cured hams [6]. Meanwhile, myosin, the most important component of myofibrils, is also a key flavor precursor of peptides and amino acids [15]. The degradation products of myosin are strongly associated with the development of taste substances of dry-cured ham. Mora et al. [16] found that myosin was closely related to the flavor formation of dry-cured hams. In our study, myosin was more abundant in LH, which is caused by the relatively higher degradation rate of structural proteins in FH, potentially contributing to the richer taste of FH. Therefore, fat removal probably had a significant effect on the protein metabolism qualities of dry-cured hams. 

To further analyze the proteins with differential abundances between FH and LH samples, hierarchical clustering analysis was performed, with the results displayed in a tree heat map in Figure 4. According to Figure 4, six FH and LH ham samples were divided into two clusters horizontally. The results showed that the triplicate samples from FH or LH were clustered into the same subset (Euclidean distance < 3.0), indicating that the differential abundance of these proteins can be attributed to the difference between the two groups during processing. Meanwhile, structural proteins such as actin-2 (F1LI18), IgG heavy chain (L8B0W4), tropomyosin 1 (A0A286XHM4), β-tropomyosin (Q8MKF3), and fibrinogen γ-chain (A0A5G2QUU1) were significantly up-regulated in FH compared with those in LH, suggesting that the lack of protection from fat and skin tissues might allow the structural proteins of LH to be more easily degraded [4]. Consistent with our findings, Lopez-Pedrouso et al. [17] found that dry-cured hams from surgically castrated pigs, commonly higher in fat content than those from immunocastrated pigs, also have more abundant structural proteins.

### 3.3. Subcellular Localization

A subcellular organelle is a micro organ with a certain morphology and function in the cytoplasm, which is an important place in which proteins have different functions. A software for predicting subcellular structure (CELLO) was used to further explore the role of DPs in cells by subcellular localization analysis [18], and the results are illustrated in Figure 5. As shown in Figure 5a, DPs of LH vs. FH were classified into eight categories based on subcellular location. These proteins mainly derived from the cytoplasm and nucleus, which accounted for 64% of the total DPs. Moreover, mitochondrial, extracellular, plasma membrane, cytoskeletal, and endoplasmic reticulum DPs accounted for 15.38%, 9.83%, 4.70%, 4.27%, and 1.28%, respectively. Interestingly, the distribution of up-regulated and down-regulated DPs from FH in some subcellular organelles was significantly different. According to Figure 5b, the different DPs in FH and LH were mainly distributed in the mitochondrion, extracellular region, and plasma membrane. Moreover, the up-regulated DPs were more distributed in the extracellular region, whereas the down-regulated DPs were more present in mitochondrion and in the plasma membrane. These results were probably attributed to oxidative phosphorylation catalyzed by ubiquinone reductase and cytochrome-c oxidase, which were up-regulated in LH samples.

### 3.4. Gene Ontology (GO) Analysis

DPs were further annotated through GO to fully understand their functions, localizations, and related biological pathways in organisms, and the results are illustrated in Figure 6. GO functional annotation was divided into three categories including biological process, molecular function, and cellular component. Meanwhile, the number of DPs was counted at the level of GO secondary functional annotation for LH vs. FH comparison. For the biological process category, the identified DPs were mainly located in cellular process (GO:0009987) and metabolic process (GO:0008152) terms. For the molecular function category, the identified DPs mainly converged in catalytic activity (GO:0003824) and binding (GO:0005488) terms. And DPs in the cellular component category mostly belonged to cell (GO:0005623), cell part (GO:0044464), organelle (GO:0043226), and organelle part (GO:0044422). Interestingly, the number of DPs located in the catalytic activity of molecular function showed a largely significant difference in LH vs. FH, among which more DPs with active catalytic ability were from LH samples.

The enrichment of GO items under the three GO categories was shown by the bubble diagrams in Figure 7. According to the cellular component annotation (Figure 7a), vimentin, lamin isoform A, and IF rod domain-containing protein were located in both intermediate filament cytoskeleton item (GO:0045111) and polymeric cytoskeletal fiber item (GO:0099513), which were down-regulated in LH vs. FH. As for biological process annotation (Figure 7b), fatty acid transport protein 1a, belonging to the acylglycerol metabolic process, triglyceride metabolic process, neutral lipid metabolic process, and glycerolipid metabolic process, was found up-regulated in LH vs. FH. However, apolipoprotein A-I, also belonging to the above four biological process items, was significantly down-regulated in LH vs. FH. Thus, fatty acid transport protein 1a and apolipoprotein A-I can be used as the biomarker proteins of fat metabolism during the ripening of FH and LH. In the organic hydroxy compound biosynthetic process, pyridoxal 5-phosphate synthase and phosphoinositide phospholipase C were up-regulated in FH, while aspartate aminotransferase was up-regulated in LH, which might contribute to the flavor formation based on hydroxy compounds. As for the positive regulation of actin filament bundle assembly, synaptopodin was up-regulated in FH, which was coupled to actin microfilaments and related to the conformational structure degradation of muscle proteins [19]. In terms of molecular function annotation, phosphoinositide phospholipase C, protein phosphatase, fructose-bisphosphatase, aminoacyl-tRNA hydrolase, and deoxyribonuclease were down-regulated in LH vs. FH, and they were all associated with the hydrolase activity (acting on ester bonds). Overall, lipid-metabolism-related GO functions were more active in FH than in LH.

### 3.5. Kyoto Encyclopedia of Genes and Genomes (KEGG) Pathway Analysis

KEGG pathway analysis can lead us to understand the specific pathways associated with DPs [20]. Hence, DPs were analyzed and annotated using the KEGG pathway database, and the results are shown in Figure 8. It is clear that KEGG pathways can be divided into seven items, among which DPs are mainly located in pathways associated with amino metabolism, carbohydrate metabolism, the circulatory system, and energy metabolism.

The KEGG pathway enrichment analysis for DPs in LH vs. FH was performed with the results illustrated in Figure 9 (bubble diagram) and Figure 10 (map). On the one hand, as described in Figure 9, fat digestion and absorption (ssc04975), complement and coagulation cascades (ssc04610), nucleotide excision repair (ssc03420), amino sugar and nucleotide sugar metabolism (ssc00520), amino acid biosynthesis and metabolism (ssc00400, ssc00360, ssc00350, ssc00250, ssc00220), and TCA cycle (ssc00020) showed the highest rich factor among all pathways. Meanwhile, cardiac muscle contraction (ssc04260), oxidative phosphorylation (ssc00190), and thermogenesis (ssc04714) showed a higher significant level of enrichment degree than other pathways. On the other hand, as displayed in Figure 10, KEGG pathways of complement and coagulation cascades (ssc04610), amino sugar and nucleotide sugar metabolism (ssc00520), TCA cycle (ssc00020), starch and sucrose metabolism (ssc00500), pentose phosphate pathway (ssc00030), and glycolysis/gluconeogenesis (ssc00010) were down-regulated in LH vs. FH. According to the DP analysis, pyruvate carboxylase and glucose-6-phosphate showed the same tendency as the TCA cycle and glycogen metabolism (Table 1 and Figure 4). However, oxidative phosphorylation (ssc00190), protein processing in endoplasmic reticulum (ssc04141), arginine, phenylalanine, tyrosine and tryptophan biosynthesis (ssc00220, ssc00250), alanine, aspartate, glutamate, tyrosine and phenylalanine metabolism (ssc00250, ssc00350, ssc00360), purine metabolism (ssc00230), and pyrimidine metabolism (ssc00240) were up-regulated in LH vs. FH. Zhou et al. [21] found that ham flavor can be affected by the DPs involved in nucleotide metabolism. Thus, purine and pyrimidine metabolism might contribute to the flavor difference between FH and LH.

### 3.6. Connections between DPs and Metabolic Pathways

Based on the DP profiles and KEGG analysis, eight metabolic pathways are summarized in Figure 11a, which could be considered as the main pathways influenced by fat tissue during ham ripening. Among the eight KEGG pathways, amino acid metabolism and biosynthesis, including arginine, aspartate and glutamate metabolism, proline metabolism, cysteine metabolism, phenylalanine metabolism, and tyrosine biosynthesis, were significantly up-regulated in LH. Meanwhile, aspartate transaminase showed the same trend, implying its probable leading role in amino acid metabolism. However, another four relevant pathways were up-regulated in FH, among which pyruvate carboxylase was vital for both the TCA cycle and pyruvate metabolism. Moreover, glucose-6-phosphate isomerase was closely connected with glycolysis/gluconeogenesis and pentose phosphate pathways. Zhang et al. [19] reported that glucophosphatase was involved in the glycolysis of muscle tissues in frozen whiteleg shrimps, which was consistent with our above finding. 

Oxidative phosphorylation, occurring in mitochondria, is an important aerobic activity in cells and the main pathway for the production of ATP in animals [22,23], wherein ATP is synthesized by the coupling reaction of ADP and inorganic phosphoric acid [24]. The protein involved in aerobic metabolism in muscle cells plays an important role in ham quality change. In addition, the texture properties of ham are also closely related to muscle structural proteins [25]. Figure 11b describes the differences between FH and LH in the oxidative phosphorylation pathway and structural protein profile. It was found that oxidative phosphorylation-related enzymes (ubiquinone reductase and cytochrome-c oxidase) were significantly up-regulated in LH. Ubiquinone reductase is an important enzyme in the mitochondrial respiratory chain, and it participates in the electron transfer reaction during oxidative phosphorylation [26]. Subcellular localization analysis showed that the number of up-regulated proteins of LH located in mitochondria was more than that of FH (Figure 5b), which was consistent with the trend of oxidative phosphorylation. For structural proteins, their abundances and degradation were closely related to the texture and taste quality of ham [23,27,28]. As shown in Figure 11b, tropomyosin and actin were up-regulated while myosin was up-regulated in LH vs. FH, implying that these DPs might contribute to different taste properties between FH and LH.

## 4. Discussion

DPs have important effects on the metabolism of nutrients, which leads to changes in the physicochemical characteristics and organoleptic properties of Jinhua hams. According to the DP profiles (Table 1 and Figure 4), a number of enzymes showed significant differences in LH vs. FH. In FH, the up-regulated enzymes, such as glucose-6-phosphate isomerase and pyruvate carboxylase, were mainly involved in glycogen metabolism. Our previous study found that the intensities of volatile flavor compounds, such as volatile alcohols, acids, and aldehydes, are significantly different in FH and LH, which was closely associated with the microbial metabolism [4]. Fat removal can significantly affect the microbial community and flavor characteristics of FH and LH. Specifically, *Ruminococcaceae* decomposed unsaturated fatty acids into linear-chain aldehydes through β-oxidation, which were further reduced to linear-chain alcohols, followed by esterification with the participation of *Debaryomyces* [4]. Consequently, FH was better able to consume pyruvate, which usually promotes glycogen metabolism (Figure 11a). Moreover, aldehydes are considered as key contributors to the distinctive flavor of dry-cured hams due to their high concentration and low aroma threshold [1]. 

However, the lack of fat and core microorganisms led to a relative enhancement of protein metabolism [4], which corresponded with our above findings (Figure 10 and Figure 11a). Without the protection of fat tissues, microorganisms might more easily affect protein degradation. Under the proteolysis of *Staphylococcus* and *Psychrobacter*, more proteins were degraded into amino acids, and then they were subjected to Strecker degradation and deamination for the further generation of some aromatic aldehydes or linear-chained alcohols [4]. Moreover, microorganisms played an important role in promoting the formation of quality characteristics of ham products [29]. The oxidative deamination and decarboxylation of valine, leucine, and isoleucine through the Strecker reaction are able to produce more branched aldehydes, which are thought to be responsible for the grassy, fatty, or nutty taste of meat products [1,4]. In addition, the present study shows that tropomyosin and actin are more easily degraded in LH, while myosin is degraded more in FH (Table 1 and Figure 4). The abundance of α-actin was found to be closely associated with beef tenderness [29], and the same result was also reported in pork [24]. More importantly, the degradation of structural proteins might lead to the characteristic taste and flavor of dry-cured hams [28,30]. The degradation mechanism of different structural proteins during the ripening of dry-cured ham should be further studied to understand its contribution to the formation of the differential flavor of FH and LH.

On the other hand, the production of ATP from glycogen oxygen metabolism is over 10-fold greater than that from anaerobic glycolysis, which implies that even a small increasement in oxidative phosphorylation can boost ATP production [31]. According to the report of Ramos et al. [32], energy metabolism is an important metabolic process that occurs in muscle tissues after animal death, and oxidative phosphorylation has an important impact on meat quality, especially on post-mortem tenderness. The level of oxidative phosphorylation was significantly up-regulated in LH (Figure 10 and Figure 11b), which could result in the production of much more ATP. Furthermore, it could also be conducive to the reabsorption of calcium by the sarcoplasmic reticulum calcium pump, delaying calpain activation, decreasing the proteolytic degree, and thus reducing the tenderness of post-mortem meats [28]. Additionally, the anti-aging factor heat shock protein was also up-regulated in LH (Table 1 and Figure 4), which might delay tenderization by regulating the activity of caspase-3 [33]. Although oxidative phosphorylation tends to delay the degree of proteolysis through energy metabolism, LH still shows a higher level of protein metabolism than FH. However, the effect and mechanism of fat removal on the texture characteristics of Jinhua ham remain to be further investigated.

## 5. Conclusions

In this study, label-free proteomics was used to explore the protein differentiation mechanism between Jinhua FH and LH. A number of DPs were significantly detected in LH vs. FH, which indicated that great variations were caused by the subcutaneous fat in the pig’s hind leg. Compared to FH, actin, tropomyosin, and aspartate aminotransferase were up-regulated in LH, while myosin, pyruvate carboxylase, and glu-cose-6-phosphate isomerase were down-regulated in LH. Bioinformatic analyses revealed that the DPs were mainly located in cellular process, metabolic process, catalytic activity, binding, cell, cell part, organelle and organelle part GO terms. From the perspective of protein differentiation, fat removal mainly caused the up-regulated KEGG pathways of amino acid metabolism, and the down-regulated KEGG pathways of the TCA cycle, pyruvate metabolism, glycolysis/gluconeogenesis, and pentose phosphate. The above findings provide the theoretical insights into the protein, glycogen, and fat metabolism of Jinhua ham between FH and LH, and are useful for understanding the effect of fat removal on the flavor and taste of ham, thus laying the foundation for the development of healthier, low-fat ham.

## Figures and Tables

**Figure 1 foods-12-04348-f001:**
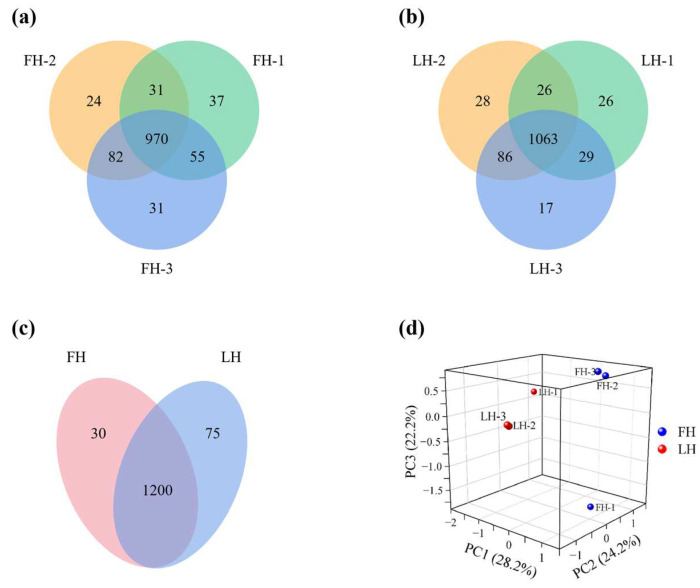
Venn diagram of proteins identified within (**a**,**b**) and between (**c**) FH and LH groups, and principal component analysis (PCA) of proteins identified between FH and LH (**d**). FH, fatty ham; LH, lean ham.

**Figure 2 foods-12-04348-f002:**
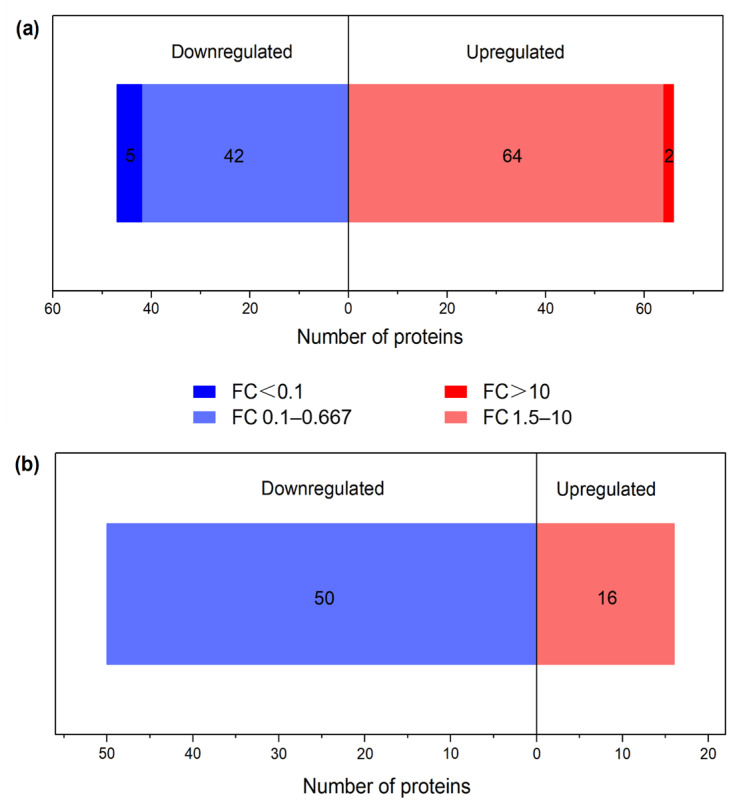
DP profiles of FH vs. LH. (**a**) Overlapping proteins with differential abundances; (**b**) different proteins with consistent presence/absence; FH, fatty ham; LH, lean ham; FC, fold change.

**Figure 3 foods-12-04348-f003:**
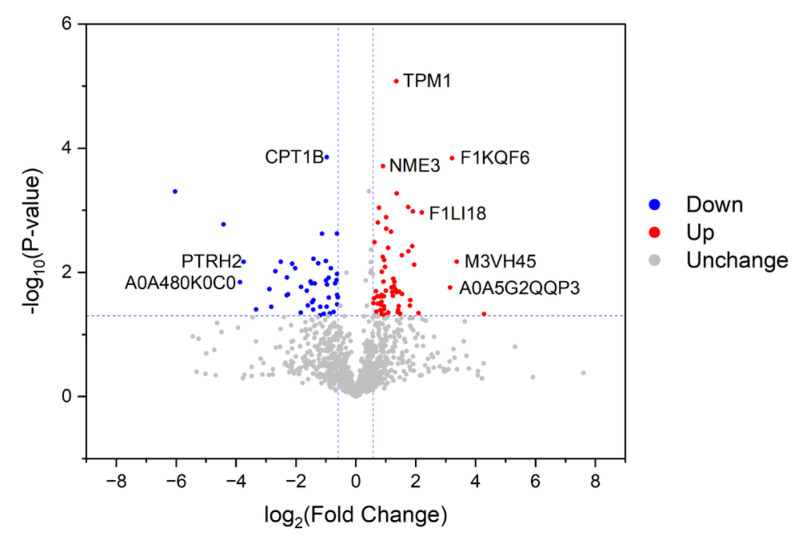
Volcano map of proteins with differential abundances between FH and LH. Significantly down-regulated (FC < 0.667 and *p* < 0.05) and up-regulated (FC >1.5 and *p* < 0.05) proteins are marked in blue and red, respectively, whereas undifferentiated proteins are marked in gray. FH, fatty ham; LH, lean ham; FC, fold change.

**Figure 4 foods-12-04348-f004:**
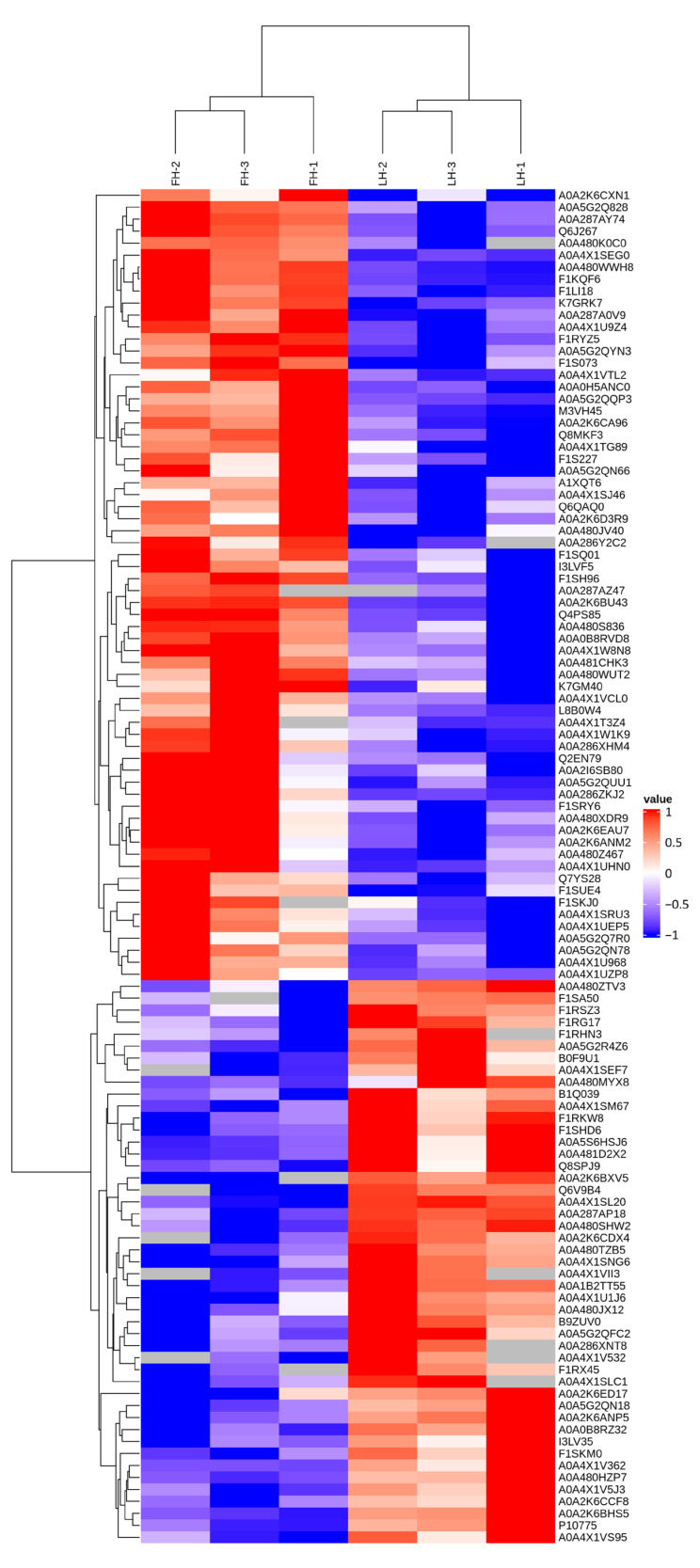
Cluster analysis of proteins with differential abundances in FH and LH. The color gradation represents the Z-score of protein abundance, and the undetected proteins are marked in gray. FH, fatty ham; LH, lean ham.

**Figure 5 foods-12-04348-f005:**
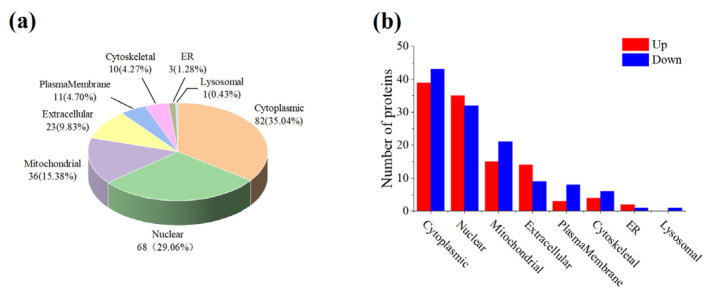
Subcellular localization of all DPs (**a**) and distributions of up- and down-regulated DPs for FH vs. LH at the subcellular level (**b**). LH, lean ham; FH, fatty ham; ER, endoplasmic reticulum.

**Figure 6 foods-12-04348-f006:**
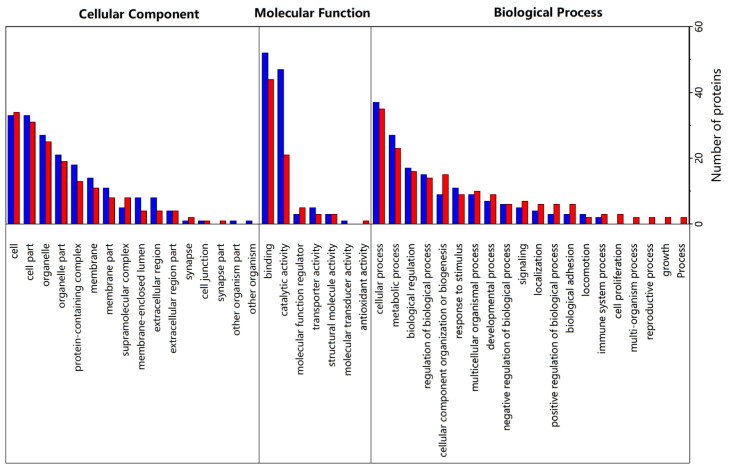
GO secondary functional annotation of DPs from LH vs. FH comparison. Significantly up-regulated and down-regulated proteins are marked in blue and red, respectively. LH, lean ham; FH, fatty ham.

**Figure 7 foods-12-04348-f007:**
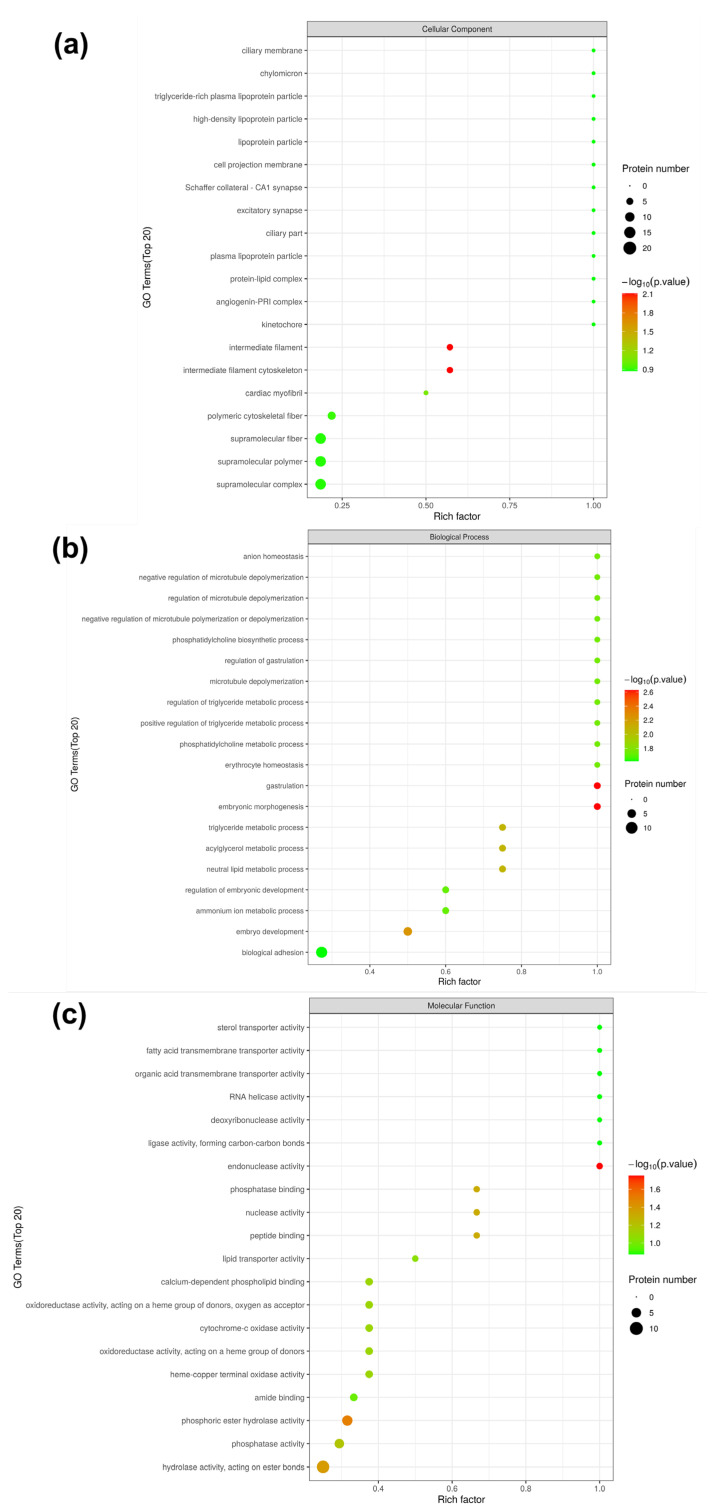
GO functional enrichment bubble map of DPs from LH vs. FH under sub-functional classification of cellular component (**a**), biological process (**b**), and molecular function (**c**). LH, lean ham; FH, fatty ham.

**Figure 8 foods-12-04348-f008:**
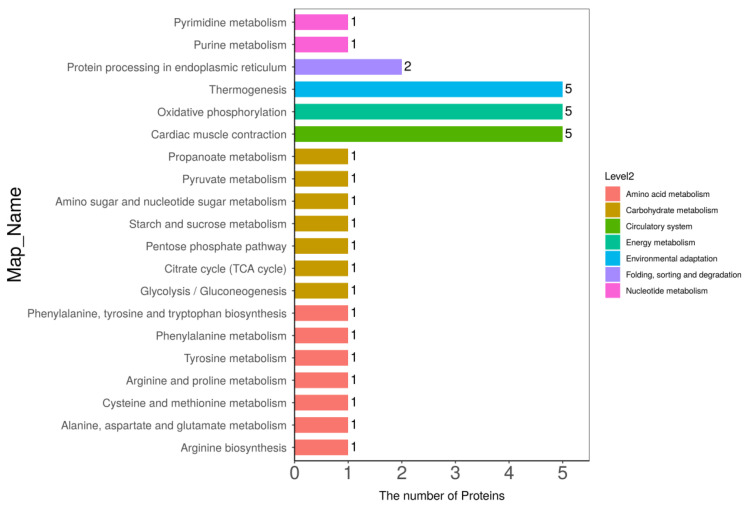
Annotation and attribution of KEGG pathways for DPs in LH vs. FH comparison. FH, fatty ham; LH, lean ham.

**Figure 9 foods-12-04348-f009:**
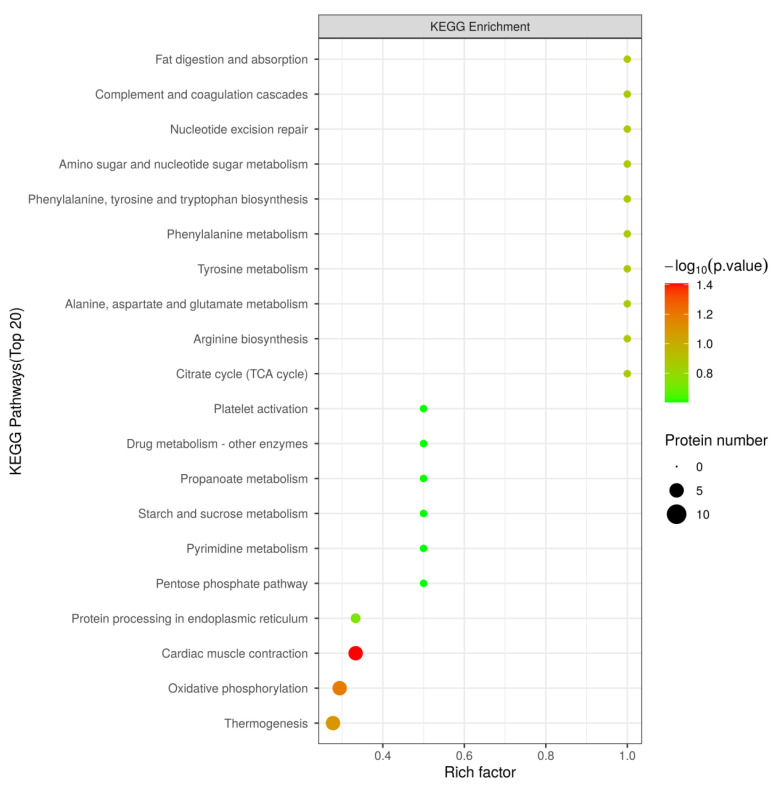
Bubble diagram of KEGG pathway enrichment (top 20) for DPs in LH vs. FH comparison. LH, lean ham; FH, fatty ham.

**Figure 10 foods-12-04348-f010:**
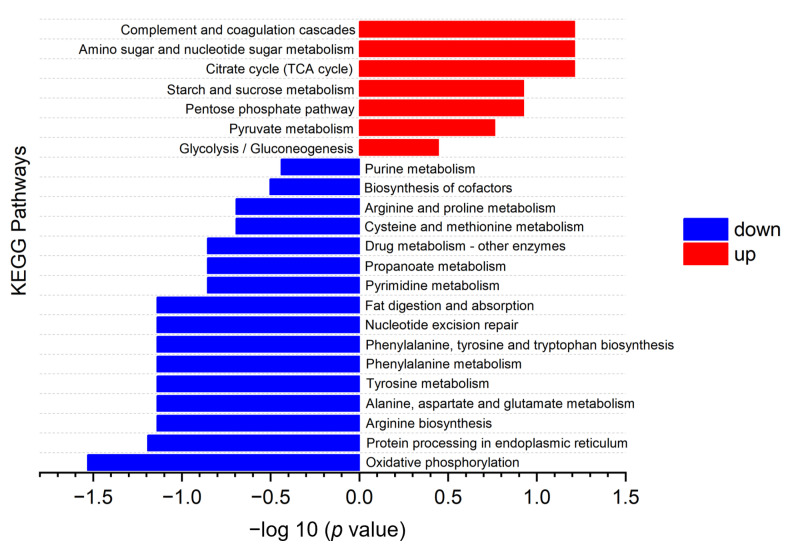
KEGG pathway enrichment map of DPs for FH vs. LH comparison. Significantly down-regulated and up-regulated proteins are marked in blue and red, respectively. LH, lean ham; FH, fatty ham.

**Figure 11 foods-12-04348-f011:**
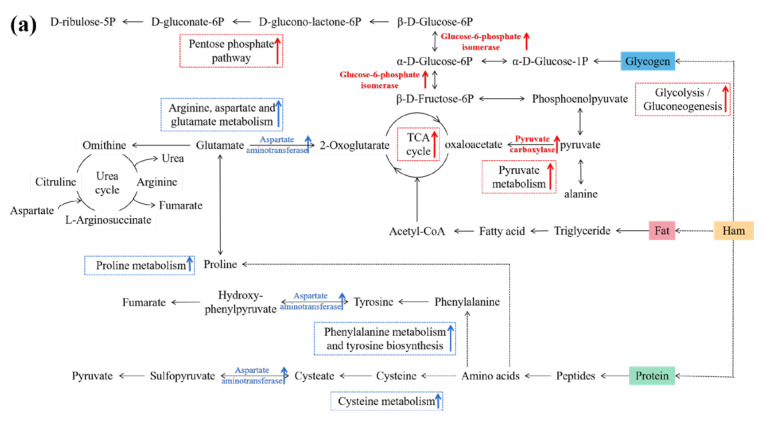
Potential metabolic pathways of nutrients (glycogen, fat, and protein) (**a**), oxidative phosphorylation, and structural proteins (**b**) based on KEGG enrichment for LH vs. FH comparison (**b**). The blue and red colors represent up-regulations of DPs and metabolic pathways in LH and FH, respectively. FH, fatty ham; LH, lean ham.

**Table 1 foods-12-04348-t001:** Characterized proteins with differential abundances (FC > 1.5 or <0.667 and *p* < 0.05) in FH and LH.

Protein ID	Gene ID	Protein Name	Intensity (Log_10_ *P*)		FC (FH/LH)	*p* Value
			FH	LH		
M3VH45	*ANXA6*	Annexin	5.95 ± 0.12	4.75 ± 0.52	10.32	0.01
F1KQF6		Glutamyl-tRNA synthetase (Fragment)	5.48 ± 0.04	4.49 ± 0.17	9.25	0.00
A0A5G2QQP3		Histone H2A	6.51 ± 0.16	5.55 ± 0.23	8.84	0.02
F1LI18		Actin-2 (Fragment)	6.50 ± 0.06	5.82 ± 0.14	4.61	0.00
L8B0W4	*IGHG*	IgG heavy chain	6.27 ± 0.18	5.64 ± 0.17	4.27	0.05
A0A286XHM4	*TPM1*	Tropomyosin 1	6.84 ± 0.10	6.23 ± 0.18	3.86	0.01
Q8MKF3	*TPM2*	Beta-tropomyosin (Fragment)	7.11 ± 0.07	6.48 ± 0.29	3.69	0.00
A0A5G2QN66	*P4HB*	Protein disulfide-isomerase	4.82 ± 0.14	4.19 ± 0.33	3.52	0.03
A0A4X1SRU3	*PSMC4*	26S proteasome AAA-ATPase subunit RPT3	5.85 ± 0.14	5.25 ± 0.31	3.49	0.03
A0A5G2QYN3	*SYNPO*	Synaptopodin	5.31 ± 0.07	4.74 ± 0.24	3.37	0.00
F1RYZ5	*RPL10A*	Ribosomal protein	5.28 ± 0.05	4.74 ± 0.15	3.36	0.00
A0A286Y2C2	*PPP3R1*	Protein phosphatase 3 regulatory subunit B, alpha	5.47 ± 0.09	5.00 ± 0.11	2.92	0.02
A0A0H5ANC0	*OGN*	Mimecan	5.70 ± 0.08	5.23 ± 0.11	2.90	0.01
Q2EN79	*UQCR10*	Ubiquinol-cytochrome c reductase complex	6.03 ± 0.18	5.58 ± 0.21	2.79	0.05
A0A5G2QN78	*GRB2*	Growth factor receptor bound protein 2	5.39 ± 0.11	4.94 ± 0.17	2.73	0.02
F1SRY6	*PLCD4*	Phosphoinositide phospholipase C	5.88 ± 0.14	5.41 ± 0.28	2.70	0.03
F1SKJ0	*ARIH2*	RBR-type E3 ubiquitin transferase	5.68 ± 0.03	5.20 ± 0.26	2.66	0.04
F1SUE4	*ASPN*	Asporin	6.12 ± 0.12	5.66 ± 0.23	2.63	0.04
Q4PS85	*MYOZ1*	Myozenin-1	7.66 ± 0.04	7.25 ± 0.07	2.58	0.00
A0A480XDR9		Biglycan (Fragment)	6.58 ± 0.11	6.15 ± 0.22	2.56	0.02
A0A4X1TG89	*ALDH6A1*	Aldedh domain-containing protein	5.36 ± 0.04	4.91 ± 0.26	2.51	0.02
A0A4X1T3Z4	*DNASE1L1*	Deoxyribonuclease	5.62 ± 0.07	5.22 ± 0.12	2.42	0.02
A0A480WUT2		Endophilin-B1 isoform 1	5.01 ± 0.07	4.60 ± 0.21	2.37	0.01
F1S227	*PTRH2*	Aminoacyl-tRNA hydrolase	5.70 ± 0.11	5.32 ± 0.16	2.36	0.02
A0A480K0C0		Collagen alpha-3(VI) chain isoform X4	5.62 ± 0.01	3.48 ± 3.02	2.32	0.02
A0A2I6SB80		Thy-1 cell surface antigen (Fragment)	5.53 ± 0.12	5.19 ± 0.17	2.13	0.04
A0A4X1U9Z4	*NPM1*	Nucleoplasmin domain-containing protein	5.09 ± 0.03	4.78 ± 0.09	2.02	0.00
A0A287AY74	*FBP2*	Fructose-bisphosphatase	7.53 ± 0.02	7.22 ± 0.09	2.02	0.00
Q7YS28		Pyruvate carboxylase	5.56 ± 0.10	5.24 ± 0.18	2.01	0.05
A0A2K6CXN1	*NEB*	SH3 domain-containing protein	5.85 ± 0.10	5.54 ± 0.14	1.96	0.05
A0A5G2Q828	*PNPO*	Pyridoxal 5-phosphate synthase	5.79 ± 0.02	5.48 ± 0.16	1.96	0.01
A0A286ZKJ2	*PFKM*	ATP-dependent 6-phosphofructokinase	7.88 ± 0.07	7.60 ± 0.01	1.92	0.01
I3LVF5	*PA2G4*	Peptidase_M24 domain-containing protein	5.32 ± 0.07	5.02 ± 0.19	1.91	0.03
F1SQ01	*PRDX4*	Peroxiredoxin-4	6.16 ± 0.05	5.85 ± 0.20	1.91	0.02
A0A0B8RVD8	*VIM*	Vimentin	6.85 ± 0.03	6.55 ± 0.17	1.90	0.01
K7GM40	*APOA1*	Apolipoprotein A-I	5.72 ± 0.07	5.41 ± 0.20	1.89	0.05
A0A480WWH8		Basement membrane-specific heparan sulfate proteoglycan core protein isoform X4	6.50 ± 0.02	6.22 ± 0.02	1.88	0.00
F1S073	*ANXA2*	Annexin	7.08 ± 0.02	6.80 ± 0.11	1.87	0.01
A0A480JV40		Lamin isoform A	6.82 ± 0.05	6.53 ± 0.15	1.86	0.02
A0A4X1UHN0		IF rod domain-containing protein	6.75 ± 0.12	6.50 ± 0.06	1.82	0.05
A0A4X1UEP5	*HSPA5*	78 kDa glucose-regulated protein	6.41 ± 0.09	6.16 ± 0.07	1.79	0.03
A0A481CHK3		Alpha-actinin-4 isoform 1	6.82 ± 0.03	6.53 ± 0.15	1.79	0.03
A0A2K6D3R9	*DES*	IF rod domain-containing protein	7.91 ± 0.10	7.67 ± 0.10	1.77	0.04
A0A5G2QUU1	*FGG*	Fibrinogen gamma chain	5.96 ± 0.09	5.72 ± 0.06	1.73	0.02
F1SH96	*ITIH1*	Inter-alpha-trypsin inhibitor heavy chain H1	5.62 ± 0.01	5.39 ± 0.06	1.71	0.00
Q6J267		Galectin (Fragment)	7.42 ± 0.03	7.20 ± 0.05	1.66	0.00
A0A480Z467		Neuroblast differentiation-associated protein AHNAK isoform X1	6.03 ± 0.08	5.81 ± 0.08	1.66	0.03
Q6QAQ0		40S ribosomal protein S20 (Fragment)	5.31 ± 0.05	5.08 ± 0.11	1.65	0.02
A0A4X1SJ46	*ANXA11*	Annexin	6.20 ± 0.09	6.00 ± 0.05	1.60	0.04
A0A480S836		Glucose-6-phosphate isomerase	7.86 ± 0.03	7.65 ± 0.12	1.59	0.02
A0A287A0V9	*PHKB*	Phosphorylase b kinase regulatory subunit	6.13 ± 0.03	5.94 ± 0.04	1.54	0.00
A1XQT6	*MYL1*	MLC1f	8.11 ± 0.06	7.93 ± 0.06	1.52	0.03
F1RX45	*DIP2C*	Disco interacting protein 2 homolog C	4.53 ± 0.06	4.71 ± 0.04	0.66	0.02
A0A480ZTV3		Endonuclease domain-containing 1 protein	5.27 ± 0.11	5.47 ± 0.02	0.65	0.02
B1Q039	*CRYAB*	Alpha(B)-crystallin	7.42 ± 0.06	7.61 ± 0.07	0.65	0.03
A0A4X1VII3	*ATP6V1B2*	ATP-synt_ab domain-containing protein	3.06 ± 0.02	4.78 ± 0.02	0.65	0.01
A0A480SHW2		Myosin-7	8.57 ± 0.06	8.76 ± 0.01	0.65	0.00
F1RKW8	*PSMD11*	Proteasome 26S subunit, non-ATPase 11	5.77 ± 0.07	5.96 ± 0.05	0.64	0.01
F1SKM0	*UQCRC1*	Ubiquinol-cytochrome c reductase core protein 1	6.76 ± 0.05	6.97 ± 0.06	0.62	0.01
A0A2K6CCF8	*UCHL3*	Ubiquitin carboxyl-terminal hydrolase	5.91 ± 0.06	6.13 ± 0.09	0.60	0.04
F1SA50	*HNRNPM*	Heterogeneous nuclear ribonucleoprotein M	4.73 ± 0.21	5.02 ± 0.01	0.55	0.05
A0A5S6HSJ6	*ATP2A2*	Calcium-transporting ATPase	7.06 ± 0.03	7.33 ± 0.08	0.53	0.01
B9ZUV0	*COII*	Cytochrome c oxidase subunit 2	6.60 ± 0.15	6.90 ± 0.06	0.51	0.02
A0A4X1SL20	*MYH2*	Myosin-2	7.92 ± 0.05	8.22 ± 0.01	0.51	0.00
F1RSZ3	*ACSL1*	Acyl-CoA synthetase long chain family member 1	6.32 ± 0.25	6.65 ± 0.05	0.51	0.04
A0A5G2QFC2	*MDH1*	Malate dehydrogenase	6.35 ± 0.12	6.66 ± 0.07	0.50	0.01
A0A287AP18	*NRAP*	Nebulin related anchoring protein	5.62 ± 0.15	5.94 ± 0.01	0.50	0.01
F1RHN3	*COMT*	Catechol O-methyltransferase	5.30 ± 0.19	5.64 ± 0.07	0.48	0.05
A0A1B2TT55	*GOT1*	Aspartate aminotransferase	7.07 ± 0.10	7.42 ± 0.04	0.46	0.00
B0F9U1		Alpha 1 S haptoglobin (Fragment)	5.04 ± 0.15	5.41 ± 0.12	0.44	0.04
A0A4X1V532	*NDUFB7*	Complex I-B18	4.39 ± 0.09	4.76 ± 0.07	0.44	0.05
A0A4X1SLC1		Protein-serine/threonine phosphatase	4.27 ± 0.12	4.65 ± 0.02	0.42	0.01
A0A4X1SM67	*MAP2K6*	Protein kinase domain-containing protein	5.64 ± 0.13	6.06 ± 0.10	0.39	0.02
A0A0B8RZ32	*STIP1*	Stress-induced phosphoprotein 1	5.41 ± 0.11	5.84 ± 0.08	0.37	0.01
I3LV35	*CPT1B*	Carnitine O-palmitoyltransferase	5.31 ± 0.14	5.74 ± 0.14	0.37	0.04
A0A480JX12		Integrin beta	5.38 ± 0.56	6.00 ± 0.06	0.36	0.03
Q8SPJ9	*COX7A1*	Cytochrome c oxidase subunit 7A1, mitochondrial	6.09 ± 0.08	6.53 ± 0.13	0.36	0.01
P10775	*RNH1*	Ribonuclease inhibitor	5.67 ± 0.10	6.12 ± 0.10	0.35	0.01
F1RG17	*NME3*	Nucleoside diphosphate kinase	4.88 ± 0.70	5.61 ± 0.07	0.32	0.02
A0A2K6CDX4	*PGAM2*	Phosphoglycerate mutase	5.62 ± 0.39	6.25 ± 0.05	0.28	0.02
A0A2K6ED17	*CAMK2D*	Calcium/calmodulin-dependent protein kinase	5.08 ± 0.66	5.97 ± 0.07	0.28	0.04
A0A4X1SNG6	*APOBEC2*	CMP/dCMP-type deaminase domain-containing protein	5.96 ± 0.29	6.63 ± 0.09	0.25	0.01
A0A5G2R4Z6	*LMCD1*	LIM and cysteine-rich domains protein 1	5.80 ± 0.17	6.45 ± 0.11	0.23	0.01
A0A480HZP7		Protein flightless-1 homolog isoform 1	4.26 ± 0.11	4.93 ± 0.15	0.21	0.02
A0A286XNT8		TPR_REGION domain-containing protein	4.40 ± 0.52	5.24 ± 0.08	0.20	0.01
F1SHD6	*EEF1B2*	Elongation factor 1-beta	4.80 ± 0.89	5.89 ± 0.11	0.18	0.01
A0A480TZB5		Heat shock 70 kDa protein 4	5.02 ± 0.35	5.91 ± 0.12	0.16	0.01
A0A480MYX8		Laminin subunit alpha-5 isoform X1	4.41 ± 0.25	5.27 ± 0.23	0.14	0.04
A0A5G2QN18	*SUGT1*	SGT1 homolog, MIS12 kinetochore complex assembly cochaperone	4.69 ± 0.84	5.89 ± 0.15	0.14	0.02
A0A481D2X2		Protein-serine/threonine kinase	3.87 ± 0.62	5.17 ± 0.19	0.07	0.01
Q6V9B4	*APM1*	Adiponectin (Fragment)	3.74 ± 0.26	5.59 ± 0.03	0.02	0.00

Uncharacterized proteins are not shown in the table. Log_10_
*P*, the logarithmic value of ion abundance in MS spectra; FH, fatty ham; LH, lean ham; FC, fold change.

## Data Availability

The data is contained within the article.

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
