# Peer review of "Insights into the Protein Differentiation Mechanism between Jinhua Fatty Ham and Lean Ham through Label-Free Proteomics"

_foods, 2023, doi:10.3390/foods12234348_

Round 1
Reviewer 1 Report
Comments and Suggestions for Authors
The manuscript, aiming to compare the protein profile of Jinhua fatty ham (FH) and lean ham (LH), harbors an original idea. The findings are interesting and well presented in the form of convincing figures. The idea is well constructed with a decent amount of literature. I have several suggestions for improvement:
1. Could you please describe Jinhua ham in a more thorough manner from ethnic food perspectives? I believe that this ham has a significant meaning for Chinese people. Is LH common to find in Chinese market? Among all Jinhua hams in Chinese market, how much (in percentage) is LH compared to FH?
2. Abstract: The sentence "Jihua lean ham (LH), a dry-cured ham removing subcutaneous fats..." does not sound correct. Please rephrase it. I would suggest "Jinhua lean ham (LH), a dry-cured ham made from defatted pork, ....".
3. Figure 1: I believe there is a mistyping of (b). Please change it into (d).
4. In some parts of the manuscript, it is confusing to understand the difference in terms of protein profile between FH and LH. Since LH is the main focus of the paper (compared to the conventional FH), I would suggest all the results to be presented as LH vs. FH (not FH vs. LH as presented in some parts of the manuscript).
5. In the discussion, please highlight the consequences of the different protein profile between FH and LH interms of several aspects, including nutrition, physicochemical characteristics, and organoleptic properties.
6. Please take a look at sentences like L436-438.
"Among the key DEPs, actin, tropomyosin and aspartate 436 aminotransferase were up-regulated in LH, while, myosin, pyruvate carboxylase, and glu- 437 cose-6-phosphate isomerase were down-regulated in FH."
Are these proteins really "up- or down-regulated"? Compared to which subject? Fresh pork? Wouldn't it be more correct to reformulate the sentence into "Compared to FH, protein x, y, and z are found in higher amount"?
Thank you.
Author Response
|
Response to Reviewer 1 Comments |
||
|
1. Summary |
|
|
|
Thank you very much for your recognition, careful review, and valuable suggestions. We have made corresponding changes point by point according to your comments. All changes were marked in red and highlighted in the re-submitted files. |
||
|
2. Questions for General Evaluation |
Reviewer’s Evaluation |
Response and Revisions |
|
Does the introduction provide sufficient background and include all relevant references? |
Can be improved |
Responsed as follows and revised in the re-submitted files |
|
Are all the cited references relevant to the research? |
Yes |
|
|
Is the research design appropriate? |
Yes |
|
|
Are the methods adequately described? |
Yes |
|
|
Are the results clearly presented? |
Yes |
|
|
Are the conclusions supported by the results? |
Can be improved |
Responsed as follows and revised in the re-submitted files |
|
3. Point-by-point response to Comments and Suggestions for Authors |
||
|
Comments 1: Could you please describe Jinhua ham in a more thorough manner from ethnic food perspectives? I believe that this ham has a significant meaning for Chinese people. Is LH common to find in Chinese market? Among all Jinhua hams in Chinese market, how much (in percentage) is LH compared to FH? |
||
|
Response 1: Thank you for your valuable suggestion. Actually, Jinhua ham has a history of over 1,200 years. Because of its bright color, unique flavor, good taste, and high nutritive value, Jinhua ham was listed as "China National Geographic Indication Products". As a traditional ethnic food in China, Jinhua ham is widely popular among consumers as a flavor enhancer, umami improver, or nutrient fortifier in dishes. According to your advice, these descriptions of Jinhua ham from the perspective of ethnic food have been added in the revised manuscript (Page 1, Lines 36-40). On the other hand, lean ham is a quite new product without large market proportion presently, but it is becoming popular among consumers due to its low-fat attribute and the increasing health concern of people. This has also been stated in the revised manuscript (Page 2, Lines 38-40) accordingly. |
||
|
Comments 2: Abstract: The sentence "Jinhua lean ham (LH), a dry-cured ham removing subcutaneous fats..." does not sound correct. Please rephrase it. I would suggest "Jinhua lean ham (LH), a dry-cured ham made from defatted pork, ....". |
||
|
Response 2: Thank you for your valuable suggestion. According to your advice, the “a dry-cured ham removing subcutaneous fats” has been changed to “a dry-cured ham made from defatted hind legs of pig” in the revised manuscript (Page 1, Line 16). |
||
|
Comments 3: Figure 1: I believe there is a mistyping of (b). Please change it into (d). |
||
|
Response 3: Thank you for your careful review. The “(b)” was changed to “(d)” in the revised manuscript (Page 4, Line 174) accordingly. |
||
|
Comments 4: In some parts of the manuscript, it is confusing to understand the difference in terms of protein profile between FH and LH. Since LH is the main focus of the paper (compared to the conventional FH), I would suggest all the results to be presented as LH vs. FH (not FH vs. LH as presented in some parts of the manuscript). |
||
|
Response 4: Thank you for your valuable suggestion. According to your advice, the “FH vs. LH” has been modified to “LH vs. FH” across the full text of the revised manuscript (Page 1, Line 21; Page 3, Line 131; Page 5, Line 192, 208; Page 10, Line 252; Page 11, Lines 272, 279; Page 12, Lines 282, 288; Page 13, Lines 295, 298, 300, 310, 320, 322; Page 14, Lines 334, 340, 345; Page 15, Line 385, 386; Page 16, Line 392; Page 17, Line 441). |
||
|
Comments 5: In the discussion, please highlight the consequences of the different protein profile between FH and LH in terms of several aspects, including nutrition, physicochemical characteristics, and organoleptic properties. |
||
|
Response 5: Thank you for your valuable suggestion. According to your valuable comment, the consequences of different protein profile between FH and LH in terms of nutrition, physicochemical characteristics, and organoleptic properties have bene added in the Discussion part of the revised manuscript (Page 16, Lines 389-390, 406-411, 416-418). |
||
|
Comments 6: Please take a look at sentences like L436-438. "Among the key DEPs, actin, tropomyosin and aspartate 436 aminotransferase were up-regulated in LH, while, myosin, pyruvate carboxylase, and glu- 437 cose-6-phosphate isomerase were down-regulated in FH." Are these proteins really "up- or down-regulated"? Compared to which subject? Fresh pork? Wouldn't it be more correct to reformulate the sentence into "Compared to FH, protein x, y, and z are found in higher amount"? |
||
|
Response 6: Thank you for your valuable comment and suggestion, these proteins were actually compared between LH and FH samples. According to your valuable advice, this sentence was corrected to “Compared to FH, actin, tropomyosin, and aspartate aminotransferase were up-regulated in LH, while myosin, pyruvate carboxylase, and glu-cose-6-phosphate isomerase were down-regulated in LH” in the revised manuscript (Page 17, Lines 442-444). |
||
|
4. Response to Comments on the Quality of English Language |
||
|
Point 1: English language fine. No issues detected. |
||
|
Response 1: Thank you for your recognition. |
||
|
5. Additional clarifications |
||
|
To Editor: Considering Honggang Tang's outstanding contribution of conceptualization, resources, investigation, and funding to the article, he is regarded as a co-first authors. |
||

Reviewer 2 Report
Comments and Suggestions for Authors
L72-74: protein expression in animal postmortem? Should it be protein metabolism? The fat was removed after the animals were killed, I am not sure using “protein expression” is appropriate here. Protein expression is used in living organisms, while the focus of your study was not on the selection of animals which may have different gene/protein expression and affect the final quality. You chose randomly three legs with and without subcutaneous fat before turning into ham, therefore, you should not comment on the differences in individual legs (i.e., different protein expression), since your hypothesis was the legs were “the same” before fat-removal. If they were different, a high number of biological replicates was required to have enough statistical power.
Also, fat deficiency? I think fat-removal will be the right term here. Please revise throughout the paper.
L81: three biological replicates were quite small, although it can be used statistically, but it is unlikely to have enough power.
L90: I don’t think this citation is necessary. Please avoid unnecessary self-citation.
L140: Is it appropriate to call it “differentially expressed proteins (DEPs)”, given the fact that your focus was not on animal science. The fat was removed after the animals were killed. The main impact was the processing, not from gene expression. Should it be differentially abundant proteins?
L201 and other sections in Results: The results section should only include results, not further discussion. You can either combine the discussion or separate them completely. Please revise.
L229-232: how do you define “highly replicative”? Figure 1 and 4 both showed that there are around 20% of proteins differed between three replicates. Do you consider the 20% a small percentage? The cluster only told you the removing of fat can lead to different protein profile, and No.1 in both groups tended to be different from other two replicates. Are those paired legs used for two treatments?
L236: why it would be easier degradation? Will this because the muscles were exposed to the heat and drying environment in LH, which could induce further degradation on the proteins? Rather than genetic expression??
L267-271: this should be in methodology section.
L365-368: you meant in the dead cells or living cells? Will these functions still be relevant in the dead cells?
L408: It is not clear the connection between lack of fat and core microorganisms and enrichment in protein metabolisms, and why this was the case.
L423: generation of more ATP in processed meat??
Comments on the Quality of English LanguageLanguage is fine.
Author Response
|
Response to Reviewer 2 Comments |
||
|
1. Summary |
|
|
|
Thank you very much for your recognition, careful review, and valuable suggestions. We have made corresponding changes point by point according to your comments. All changes were marked in red and highlighted in the re-submitted files. |
||
|
2. Questions for General Evaluation |
Reviewer’s Evaluation |
Response and Revisions |
|
Does the introduction provide sufficient background and include all relevant references? |
Yes |
|
|
Are all the cited references relevant to the research? |
Can be improved |
Responsed as follows and revised in the re-submitted files |
|
Is the research design appropriate? |
Can be improved |
Responsed as follows and revised in the re-submitted files |
|
Are the methods adequately described? |
Yes |
|
|
Are the results clearly presented? |
Yes |
|
|
Are the conclusions supported by the results? |
Yes |
|
|
3. Point-by-point response to Comments and Suggestions for Authors |
||
|
Comments 1: L72-74: protein expression in animal postmortem? Should it be protein metabolism? The fat was removed after the animals were killed, I am not sure using “protein expression” is appropriate here. Protein expression is used in living organisms, while the focus of your study was not on the selection of animals which may have different gene/protein expression and affect the final quality. You chose randomly three legs with and without subcutaneous fat before turning into ham, therefore, you should not comment on the differences in individual legs (i.e., different protein expression), since your hypothesis was the legs were “the same” before fat-removal. If they were different, a high number of biological replicates was required to have enough statistical power. Also, fat deficiency? I think fat-removal will be the right term here. Please revise throughout the paper. |
||
|
Response 1: Thank you for your careful review and valuable comments. Firstly, we agreed that the “protein expression” was not much appropriate in this study. According to your valuable comment, the “differentially expressed proteins (DEPs)” has been corrected to “differential proteins (DPs)” across the full text in the revised manuscript (Page 1, Line 20; Page 4, Lines 149, 165; etc.). Secondly, we also agreed that the analysis of DPs should be focused on the differences between LH and FH rather than the differences among individual legs. According to your valuable suggestion, the descriptions about differences in individual legs have been removed in the revised manuscript (Page 4, Lines 167-169). Thirdly, according to your valuable suggestion, the “fat deficiency” has been changed to “fat-removal” across the full text in the revised manuscript (Page 1, Lines 28, 30; Page 2, Lines 53-54, 76-77; Page 17, Line 447; etc.). |
||
|
Comments 2: L81: three biological replicates were quite small, although it can be used statistically, but it is unlikely to have enough power. |
||
|
Response 2: Thank you for your careful review. According to many previously peer-reviewed publications [1-6], three biological replicates are enough for the evaluation of proteomics, and reliable analysis results can be obtained from three biological replicates (Page 4, Lines 167-169). Some of the references using three biological replicates for proteomics study of dry-cured ham and its associated meat products are shown as follows: [1] Zhou, C.-Y., Xia, Q., He, J., Sun, Y.-Y., Dang, Y.-L., Zhou, G.-H., Geng, F., Pan, D.-D., and Cao, J.-X., Insights into ultrasonic treatment on the mechanism of proteolysis and taste improvement of defective dry-cured ham. Food Chemistry, 2022. 388.10.1016/j.foodchem.2022.133059. [2] Zhu, N., Wang, S.-w., Zhao, B., Zhang, S.-l., Zang, M.-w., Wu, Q.-r., Li, S., and Qiao, X.-l., Label-free proteomic strategy to identify proteins associated with quality properties in sauced beef processing. Food Bioscience, 2021. 42.10.1016/j.fbio.2021.101163. [3] Weng, K., Huo, W., Gu, T., Bao, Q., Cao, Z., Zhang, Y., Zhang, Y., Xu, Q., and Chen, G., Quantitative phosphoproteomic analysis unveil the effect of marketable ages on meat quality in geese. Food Chem, 2021. 361, 130093.10.1016/j.foodchem.2021.130093. [4] Zhou, C.-Y., Wu, J.-Q., Tang, C.-B., Li, G., Dai, C., Bai, Y., Li, C.-B., Xu, X.-L., Zhou, G.-H., and Cao, J.-X., Comparing the proteomic profile of proteins and the sensory characteristics in Jinhua ham with different processing procedures. Food Control, 2019. 106.10.1016/j.foodcont.2019.06.020. [5] Zhou, C.Y., Wang, C., Tang, C.B., Dai, C., Bai, Y., Yu, X.B., Li, C.B., Xu, X.L., Zhou, G.H., and Cao, J.X., Label-free proteomics reveals the mechanism of bitterness and adhesiveness in Jinhua ham. Food Chem, 2019. 297, 125012.10.1016/j.foodchem.2019.125012. [6] Mi, S., Li, X., Zhang, C.H., Liu, J.Q., and Huang, D.Q., Characterization and discrimination of Tibetan and Duroc x (Landrace x Yorkshire) pork using label-free quantitative proteomics analysis. Food Res Int, 2019. 119, 426-435.10.1016/j.foodres.2019.02.016. |
||
|
Comments 3: L90: I don’t think this citation is necessary. Please avoid unnecessary self-citation. |
||
|
Response 3: Thank you for your careful review and valuable suggestion. According to your valuable advice, this unnecessary self-citation has been removed in the revised manuscript (Page 2, Lines 93-95). |
||
|
Comments 4: L140: Is it appropriate to call it “differentially expressed proteins (DEPs)”, given the fact that your focus was not on animal science. The fat was removed after the animals were killed. The main impact was the processing, not from gene expression. Should it be differentially abundant proteins? |
||
|
Response 4: Thank you for your careful review and valuable comment. we agreed that the “protein expression” was not much appropriate in this study. According to your valuable comment, the “differentially expressed proteins (DEPs)” has been corrected to “differential proteins (DPs)” across the full text in the revised manuscript (Page 1, Line 20; Page 4, Lines 149, 165; etc.). |
||
|
Comments 5: L201 and other sections in Results: The results section should only include results, not further discussion. You can either combine the discussion or separate them completely. Please revise. |
||
|
Response 5: Thank you for your valuable comments. According to your valuable suggestion, the Results section was simplified to remove further discussion in the revised manuscript (Page 15, Line 364-381). |
||
|
Comments 6: L229-232: how do you define “highly replicative”? Figure 1 and 4 both showed that there are around 20% of proteins differed between three replicates. Do you consider the 20% a small percentage? The cluster only told you the removing of fat can lead to different protein profile, and No.1 in both groups tended to be different from other two replicates. Are those paired legs used for two treatments? |
||
|
Response 6: Thanks for your careful review and valuable comments. Firstly, we agreed that “highly replicative” was not accurate herein, and hence it has been removed in the revised manuscript (Page 4, Lines 167-169). Secondly, according to many previously peer-reviewed publications [1-6], three biological replicates are enough for the evaluation of proteomics, and reliable analysis results can be obtained even with 15-20% proteins differed among replicates. Some of the references using three biological replicates for proteomics study of dry-cured ham and its associated meat products are shown as follows: [1] Zhou, C.-Y., Xia, Q., He, J., Sun, Y.-Y., Dang, Y.-L., Zhou, G.-H., Geng, F., Pan, D.-D., and Cao, J.-X., Insights into ultrasonic treatment on the mechanism of proteolysis and taste improvement of defective dry-cured ham. Food Chemistry, 2022. 388.10.1016/j.foodchem.2022.133059. [2] Zhu, N., Wang, S.-w., Zhao, B., Zhang, S.-l., Zang, M.-w., Wu, Q.-r., Li, S., and Qiao, X.-l., Label-free proteomic strategy to identify proteins associated with quality properties in sauced beef processing. Food Bioscience, 2021. 42.10.1016/j.fbio.2021.101163. [3] Weng, K., Huo, W., Gu, T., Bao, Q., Cao, Z., Zhang, Y., Zhang, Y., Xu, Q., and Chen, G., Quantitative phosphoproteomic analysis unveil the effect of marketable ages on meat quality in geese. Food Chem, 2021. 361, 130093.10.1016/j.foodchem.2021.130093. [4] Zhou, C.-Y., Wu, J.-Q., Tang, C.-B., Li, G., Dai, C., Bai, Y., Li, C.-B., Xu, X.-L., Zhou, G.-H., and Cao, J.-X., Comparing the proteomic profile of proteins and the sensory characteristics in Jinhua ham with different processing procedures. Food Control, 2019. 106.10.1016/j.foodcont.2019.06.020. [5] Zhou, C.Y., Wang, C., Tang, C.B., Dai, C., Bai, Y., Yu, X.B., Li, C.B., Xu, X.L., Zhou, G.H., and Cao, J.X., Label-free proteomics reveals the mechanism of bitterness and adhesiveness in Jinhua ham. Food Chem, 2019. 297, 125012.10.1016/j.foodchem.2019.125012. [6] Mi, S., Li, X., Zhang, C.H., Liu, J.Q., and Huang, D.Q., Characterization and discrimination of Tibetan and Duroc x (Landrace x Yorkshire) pork using label-free quantitative proteomics analysis. Food Res Int, 2019. 119, 426-435.10.1016/j.foodres.2019.02.016. Thirdly, even though the No.1 in both groups tended to be different from other two replicates, its Euclidean distance with No.2 and No.3 in the cluster tree was still very small (< 3.0), which indicated that Nos. 1-3 were still clustered into one group. According to your valuable comments, the statement “the triplicate samples from FH or LH were clustered into the same subset (Euclidean distance < 3.0)” has been added in the revised manuscript (Page 9, Line 232-233). |
||
|
Comments 7: L236: why it would be easier degradation? Will this because the muscles were exposed to the heat and drying environment in LH, which could induce further degradation on the proteins? Rather than genetic expression?? |
||
|
Response 7: Thank you for your careful review. Combined with the findings of Zhang et al. [1], we speculated that the lack of protection by fat and skin tissues in LH might allow the easier degradation of structural proteins. Accordingly, this statement has been modified to “suggesting that the lack of protection by fat and skin tissues might allow the structural proteins of LH easier to be degraded” (Page 9, Line 238-239). Reference: [1] Zhang, J., Zhao, K., Li, H., Li, S., Xu, W., Chen, L., Xie, J., and Tang, H., Physicochemical property, volatile flavor quality, and microbial community composition of Jinhua fatty ham and lean ham: A comparative study. Front Microbiol, 2023. 14, 1124770.10.3389/fmicb.2023.1124770. |
||
|
Comments 8: L267-271: this should be in methodology section. |
||
|
Response 8: Thank you for your valuable suggestion. According to your advice, this sentence has been moved to the section “2.5. Bioinformatic analysis” in the revised manuscript (Page 3, Line 143-145). |
||
|
Comments 9: L365-368: you meant in the dead cells or living cells? Will these functions still be relevant in the dead cells? |
||
|
Response 9: Thank you for your careful review and valuable comment. The analysis was based on postmortem pork. According to the report of Ramos et al. [1], energy metabolism is an important metabolic process occurred in muscle tissues after animal death. Hence, energy metabolism might still be active during meat fermentation. And during the post-ripening, we speculated the energy metabolism might be caused by the microorganisms, however, it should be further studied. Reference: [1] Ramos, P.M., Wright, S.A., Delgado, E.F., van Santen, E., Johnson, D.D., Scheffler, J.M., Elzo, M.A., Carr, C.C., and Scheffler, T.L., Resistance to pH decline and slower calpain-1 autolysis are associated with higher energy availability early postmortem in Bos taurus indicus cattle. Meat Sci, 2020. 159, 107925.10.1016/j.meatsci.2019.107925. |
||
|
Comments 10: It is not clear the connection between lack of fat and core microorganisms and enrichment in protein metabolisms, and why this was the case. |
||
|
Response 10: Thank you for your valuable comments. According to your valuable comment, the connections and consequences among fat-removal, core microorganisms, and protein metabolisms have been added in the Discussion part of the revised manuscript (Page 16, Lines 389-390, 406-411, 416-418). |
||
|
Comments 11: L423: generation of more ATP in processed meat?? |
||
|
Response 11: Thank you for your careful review. According to the report of Ramos et al. [1], energy metabolism is an important metabolic process occurred in muscle tissues after animal death. Hence, energy metabolism might still be active during meat fermentation. And during the post-ripening, we speculated that energy metabolism might be caused by the microorganisms, however, it should be further studied. Reference: [1] Ramos, P.M., Wright, S.A., Delgado, E.F., van Santen, E., Johnson, D.D., Scheffler, J.M., Elzo, M.A., Carr, C.C., and Scheffler, T.L., Resistance to pH decline and slower calpain-1 autolysis are associated with higher energy availability early postmortem in Bos taurus indicus cattle. Meat Sci, 2020. 159, 107925.10.1016/j.meatsci.2019.107925. |
||
|
4. Response to Comments on the Quality of English Language |
||
|
Point 1: Language is fine. |
||
|
Response 1: Thank you for your recognition. |
||
|
5. Additional clarifications |
||
|
To Editor: Considering Honggang Tang's outstanding contribution of conceptualization, resources, investigation, and funding to the article, he is regarded as a co-first authors. |
||
